# BootCellNet, a resampling-based procedure, promotes unsupervised identification of cell populations via robust inference of gene regulatory networks

**Yutaro Kumagai**[ORCID]*

Cellular and Molecular Biotechnology Research Institute, National Institute of Advanced Industrial Science and Technology, Higashi, Tsukuba, Ibaraki, Japan

* yutaro.kumagai@aist.go.jp

**Data Availability Statement:** All the data used in the current study are deposited in public databases or available from private company(https://cf. 10xgenomics.com/samples/cell-exp/3.1.0/manual_

## Abstract

Recent advances in measurement technologies, particularly single-cell RNA sequencing (scRNA-seq), have revolutionized our ability to acquire large amounts of omics-level data on cellular states. As measurement techniques evolve, there has been an increasing need for data analysis methodologies, especially those focused on cell-type identification and inference of gene regulatory networks (GRNs). We have developed a new method named BootCellNet, which employs smoothing and resampling to infer GRNs. Using the inferred GRNs, BootCellNet further infers the minimum dominating set (MDS), a set of genes that determines the dynamics of the entire network. We have demonstrated that BootCellNet robustly infers GRNs and their MDSs from scRNA-seq data and facilitates unsupervised identification of cell clusters using scRNA-seq datasets of peripheral blood mononuclear cells and hematopoiesis. It has also identified COVID-19 patient-specific cells and their potential regulatory transcription factors. BootCellNet not only identifies cell types in an unsupervised and explainable way but also provides insights into the characteristics of identified cell types through the inference of GRNs and MDS.

## Author summary

Single-cell omics technologies, such as single-cell RNA-seq (scRNA-seq), are instrumental in identifying novel cell subsets that are involved in various biological processes and diseases. These technologies, however, require further development in data analysis, especially in areas focused on cell-type identification and inference of interactions between genes. The problem of cell-type identification essentially involves clustering, which necessitates a balance between distinguishing different cell states and grouping similar ones together. Current clustering methods still suffer from uncertainty in determining the appropriate number of clusters and in explaining why some cells are clustered together but others are separated. The inference of interactions between genes, gene regulatory network (GRN), remains challenging due to the noisy nature of scRNA-seq. We have

**Funding:** This research has partly been supported
by Kakenhi 22H04925 (PAGS) and 23H04486
(both to YK), and by a Grant from International
Joint Usage/Research Center, the Institute of
Medical Science, the University of Tokyo (to YK).
The funders had no role in study design data
collection and analysis, decision to publish, or
preparation of the manuscript.

**Competing interests:** The authors have declared
that no competing interests exist.

developed BootCellNet, a method that infers GRNs and a set of genes dominating the network dynamics and utilizes the set to cluster cells and identify cell types. The method addresses challenges in GRN identification and clustering methods simultaneously and will facilitate the generation of the working hypotheses from a large amount of scRNA-seq data.

## Introduction

Recent advances in measurement technologies have revolutionized our ability to acquire large amounts of omics-level data. Among these technologies, single-cell RNA sequencing (scRNA-seq) has become particularly widespread. The technique allows for an unbiased investigation of the distribution and status of various cell types within a sample, providing insights into the dynamic and heterogeneous nature of cells.

As measurement techniques evolve, there has been an increasing need for data analysis methodologies[1]. One significant challenge is the definition of cell types and states from scRNA-seq data. This requires a balance between distinguishing different cell states and grouping similar ones together. In this regard, the problem essentially boils down to one of clustering. In the analysis of scRNA-seq, cells are clustered in an unsupervised manner by using various algorithms. Among those, $k$-means clustering, hierarchical clustering, spectral clustering, and Louvain clustering are examples of methods utilized frequently. However, these methods do not provide an answer for the appropriate number of clusters and an explanation of why some cells are clustered together, but others are separated, leaving uncertainty in the analysis. An alternative to unsupervised clustering is the use of supervised methods. Typically, cells are labeled by using an annotated dataset. However, this approach faces difficulties in identifying novel cell types not included in the given dataset.

The inference of gene regulatory networks (GRNs) from transcriptomic data like scRNA-seq represents another major challenge. Successful unraveling of the complex interactions and regulatory mechanisms that control gene expression within cells will offer profound insights into cellular functions and disease mechanisms. Numerous studies are dedicated to this field, employing a range of computational approaches[2]. Despite the efforts, GRN inference is still challenging, due to the noisy nature of scRNA-seq, characterized by dropouts and strong batch effects.

Once a GRN has been inferred, an essential next step is identifying genes that are crucial for the overall control and regulation of the network. Understanding which genes have a significant influence on the network can offer critical insights into the underlying biological processes and potential targets for therapeutic intervention[3]. There are several methods to explore these pivotal elements within GRNs, such as the selection of genes based on network centrality measures[4]. Among them, one of the promising approaches relies on the concept of the minimum dominating set (MDS).

The MDS is a graph theoretical concept, defined as the smallest subset of vertices in a graph such that every other vertex is adjacent to at least one vertex in this subset. MDS plays a crucial role in controlling complex networks[5]. In the context of GRNs, this translates to a minimal set of genes whose regulatory influence extends across the entire network. In GRNs, MDS can be used to identify master regulatory genes[6]. In protein interaction networks, MDS proteins are found to be enriched with essential, cancer-related, and virus-targeted genes, showing a higher impact on network resilience than hub proteins[7].

In this study, we introduce BootCellNet, a method that employs smoothing and resampling procedures to reduce the noise in scRNA-seq data. We showed that this approach robustly

infers GRN and MDS. Since GRN controls the whole transcriptome by definition, and MDS, in theory, governs the dynamics of the whole GRN, the expression pattern of genes in MDS may reflect the entire transcriptome. Leveraging this premise we utilized MDS for cell clustering. Because clusters are given based on the MDS, the method gives a rational explanation of why some cells are clustered together while others are not. Applying our method to multiple scRNA-seq datasets confirmed its effectiveness. This addresses the aforementioned challenges in GRN identification and clustering methods simultaneously.

This paper is organized as follows. In the first three sections, we introduced the BootCellNet procedure and applied it to scRNA-seq data of peripheral blood mononuclear cells. The first two sections showed the necessity of steps in BootCellNet for robust GRN inference and MDS computation. The following section showed that robustly computed MDS offers consistent clustering results, and the result was compared to the results of supervised labeling and unsupervised clustering. In the next section, we applied the method to scRNA-seq data of hematopoiesis to show if the method can be applied to continuous spectra of cell states and help to infer developmental course. The following section is devoted to the introduction of BootCellNet2, in which the unique set of control nodes is used instead of MDS, and the clustering is automated. We showed that the use of the new set of nodes and the automated clustering provides more robust clustering results. Validation using CITE-seq data was also performed in the following section to compare to unsupervised clustering. Finally, we applied BootCellNet2 to scRNA-seq data of cells from COVID-19 patients. In each application, the relevance of clustering results was evaluated by its robustness, and the biological relevance of the obtained clusters was evaluated through the expert-based labeling of the obtained clusters.

## Results

### BootCellNet enables robust GRN inference

One of the problems in GRN inference from scRNA-seq data is its noisy and unstable nature, making the inferred GRN unreliable. To overcome the noise, BootCellNet employs two procedures, smoothing of scRNA-seq data and resampling-shuffling of data to estimate $p$-values of genes and their regulations, as detailed in Methods (Fig 1).

To confirm the importance of each step, we first compared GRNs constructed with or without smoothing. GRNs were inferred 10 times each with or without smoothing, and appearances of genes/nodes and regulations/edges were counted (Fig 2A). As shown in Fig 2B, the number of appeared genes was almost the same in GRNs with smoothing as in those without smoothing before selection using $q$-values. After the selection of genes using $q$-values, however, the number of genes was higher in GRNs with smoothing. This result indicates that the smoothing procedure promotes the selection of consistent genes and stable inference of GRN.

The above result shown in Fig 2B also implies that the smoothing alone is sufficiently effective in extracting reliable genes/nodes because the resampling-based selection of significant genes/nodes did not alter the number of genes/nodes drastically. We examined the regulations/edges in the networks with or without resampling-based selection. As shown in Fig 2C, GRNs with a resampling-based selection have a higher burden of regulations/edges that appeared 10 times, and a lower burden of those that appeared only once, indicating that the selection efficiently enriches regulations/edges with high consistency.

### BootCellNet enables robust MDS inference

Once we have robustly inferred GRN, we further infer how the GRN regulates gene expression by identifying MDS in the GRN. We implemented exact MDS computation based on an integer programming formulation. GRNs inferred above were subjected to the computation of

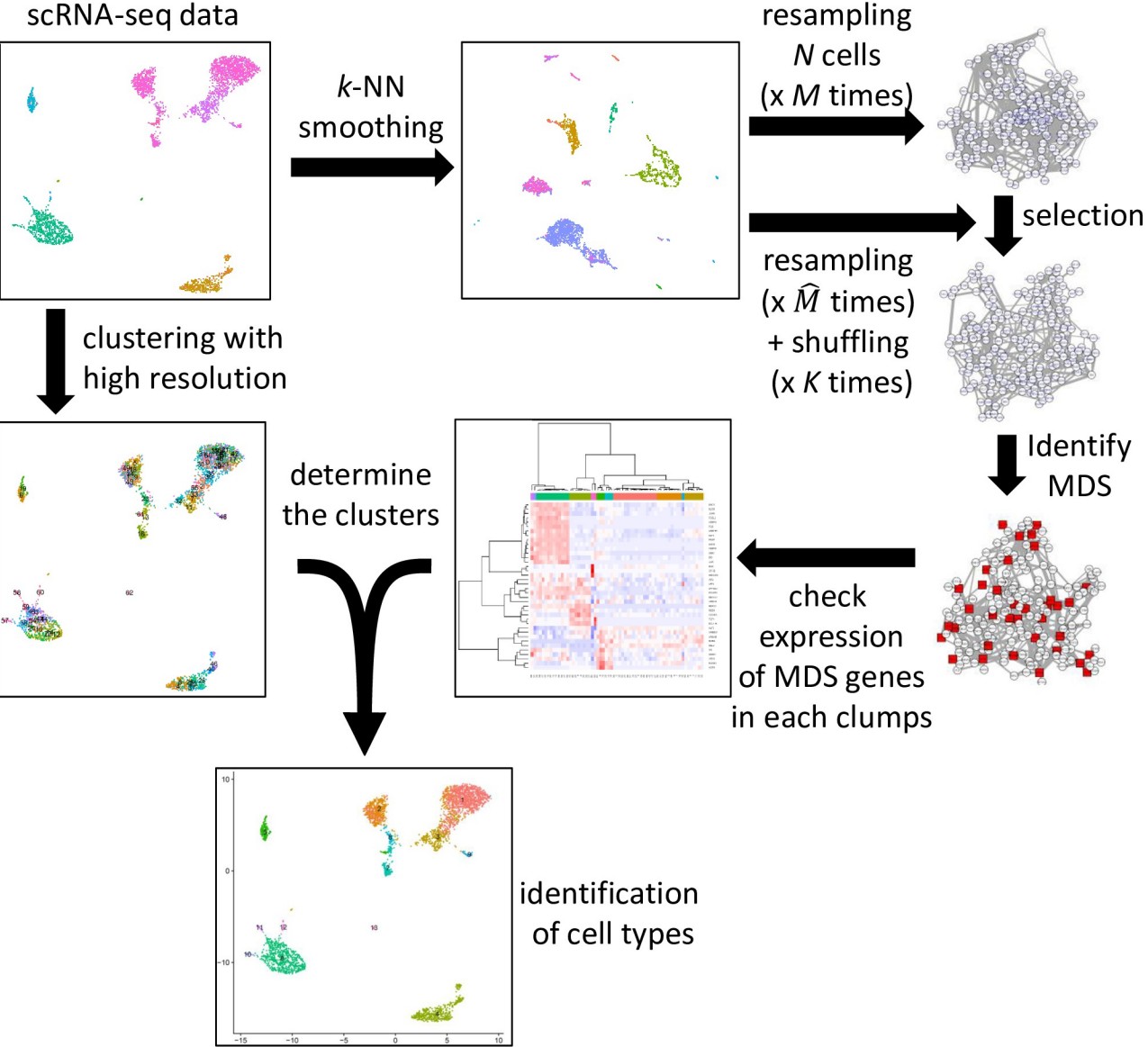

**Fig 1. The BootCellNet procedure.** Starting from scRNA-seq data (top left), $k$-NN smoothing is carried out to reduce noise. From the smoothed data, $N$ cells are sampled and GRNs are computed $M$ times. Parallelly, sampling and GRN construction are performed $\hat{M}$ times with shuffling gene names $K$ times, resulting in $\hat{M}K$ shuffled specimens. Using the shuffled data, statistically significant genes and their regulatory relationships are selected. The resulting GRN is subjected to the computation of MDS. At the same time, cells in the original scRNA-seq data are clustered into small clumps by using Louvain clustering with a high resolution. Averaged expression levels of genes in MDS are calculated and used for the clustering of cell clumps. The resulting clusters are regarded as cell types.

MDS, and then the appearance of genes in each MDS was counted. As shown in Fig 3A, the resampling-based selection increased the number of genes in MDS that appeared in all 10 specimens. Moreover, the sizes of MDS in GRNs with selection were larger than those without selection (Fig 3B), despite the selection having reduced the number of genes/nodes and regulations/edges. Furthermore, in the GRNs with the selection, MDS genes with a low degree rank appeared more consistently than in the GRNs without the selection (Fig 3C). These results indicate that the resampling-based selection of nodes and edges leads to consistent GRN structures and robust inference of MDS.

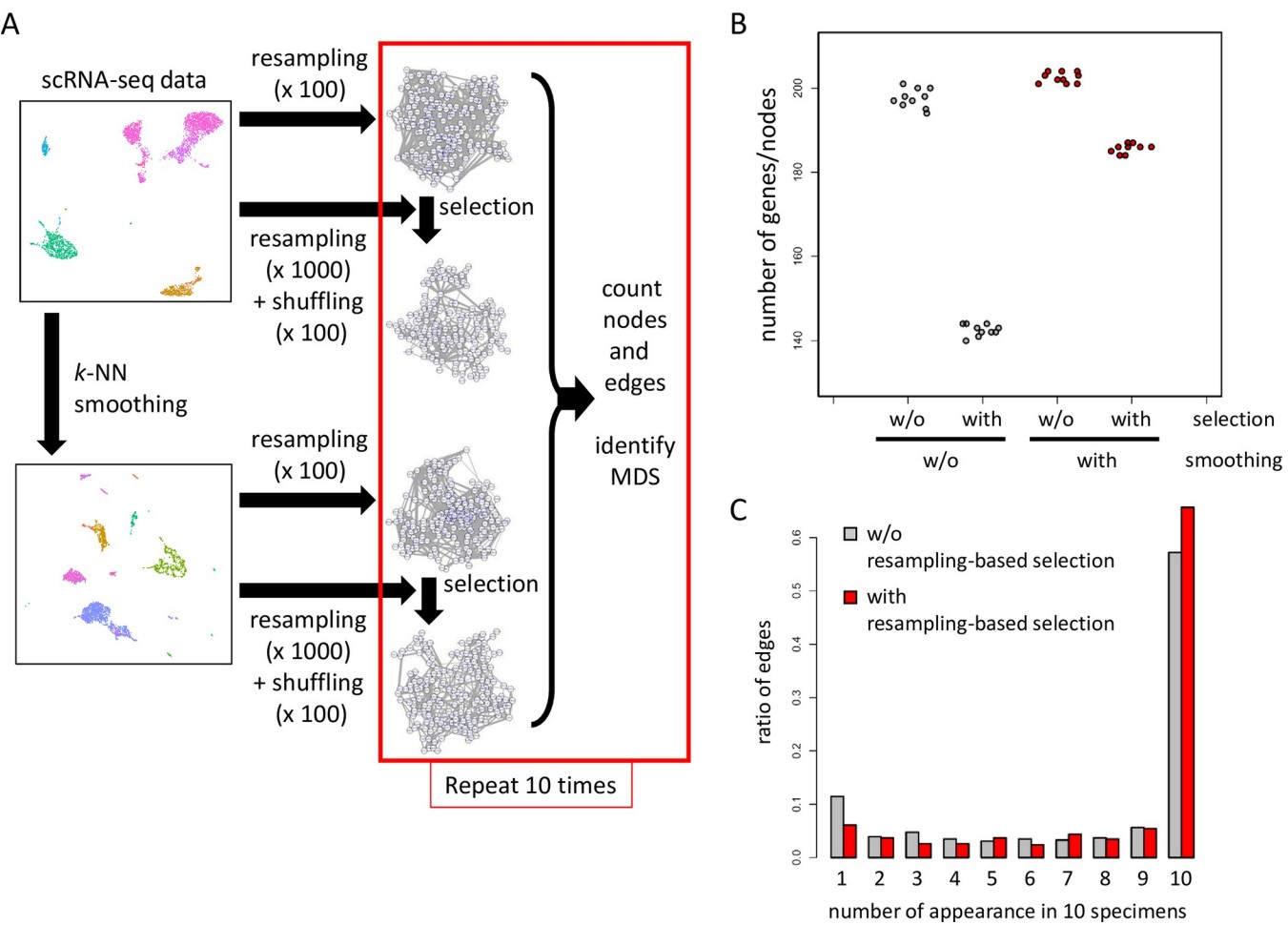

**Fig 2. Smoothing and resampling in BootCellNet are important for the robust selection of genes and regulations.** (A) Schematics of the computer experiment. (B) Numbers of genes/nodes in GRNs generated 10 times with or without smoothing and with or without selection. (C) Distribution of number of appearances of edges in 10 specimens, with (red) or without (gray) resampling-based selection.

## BootCellNet determines cell type clusters in combination with the computation of MDS

Since MDS, in theory, controls the dynamics of the whole network, the expression pattern of genes in MDS may represent the whole transcriptome. Thus, it is assumed that the expression pattern of genes in MDS can classify cells. To answer if GRN and MDS that are robustly inferred by the BootCellNet can facilitate rational cell type classification, we tested the idea on the PBMC scRNA-seq dataset. We first obtained small clusters of cells, by using Louvain algorithm with high resolution. The resolution of 10 gave 63 clusters of cells (Fig 4A). Hereafter, we call these original clusters "clumps" to distinguish them from clusters obtained later.

Then we constructed GRN and MDS was computed as shown in Fig 4B. We calculated the average expression of MDS genes in each cell clump and clustered the cell clumps as well as the genes by using hierarchical clustering with Ward's method (Fig 4C, left). By visual inspection of the heatmap of the MDS expression, we determine the number of clusters of clumps as 13 (Fig 4C, right). As a comparison, we also carried out the same by using a set of genes with variable expression instead of MDS (Fig 4D). Of 2000 variable genes, we sampled 100 genes (around three times the number of MDS, containing 36.0 genes on average; Fig 3B) and

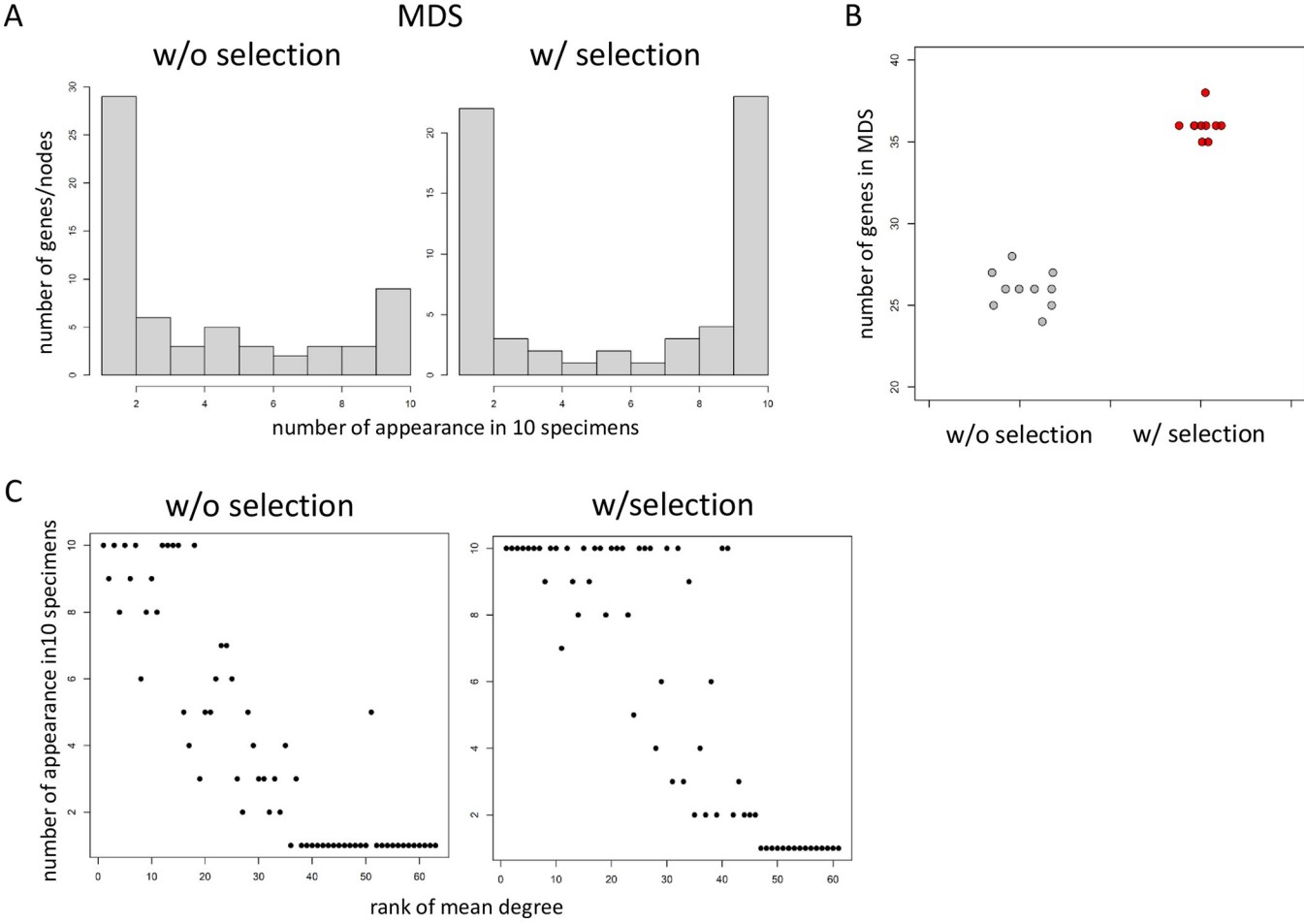

**Fig 3. BootCellNet robustly identifies MDS.** (A) Distribution of the number of the appearance of genes/nodes in 10 MDS specimens. (B) Numbers of genes in MDS with (red) or without (gray) selection. (C) Relationship between rank of mean degree and number of appearances in 10 MDS specimens of each gene.

performed the same procedure as for MDS. We then applied the number of clusters of clumps as 13, as in the procedure for MDS. The clustering was compared with each other and adjusted Rand indexes (ARIs) were calculated. As shown in Fig 4E, MDS-based clustering had higher ARIs among the 10 specimens than clustering based on 100 variable genes, indicating the robustness of MDS-based clustering.

To evaluate the obtained clusters, we picked one specimen shown in Fig 5A and checked the marker genes, Gene Ontology (GO) terms, and enriched TFBS in each cluster in the specimen. Marker genes, GO terms, and enriched TFBSs are listed in S1, S2 and S3 Tables, respectively. Cluster 1, 2, 3, and 9 exhibited characteristics of T cells. Cluster 1 is associated with the signature of CD4 T cells, whereas Cluster 2 is associated with that of CD8 T cells. Clusters 3 and 7 exhibit characteristics of activated CD4 and CD8 T cells, respectively. Cluster 8 is also associated with T cells, and based on the MDS expression pattern, it exhibits high RUNX3 expression, indicating the cluster is CD8 T cells[8]. Cluster 5 is associated with NK cell-like signatures. Cluster 4 shows B cell characteristics, marked by CD19 and immunoglobulin genes. Clusters 6, 10, and 11 feature markers whose promoters are enriched with PU.1/SPI1 transcription factor binding site (TFBS), suggesting that these clusters represent myeloid cell populations. CLEC10A and CD1C genes emerged as markers of Cluster 10, pointing to myeloid

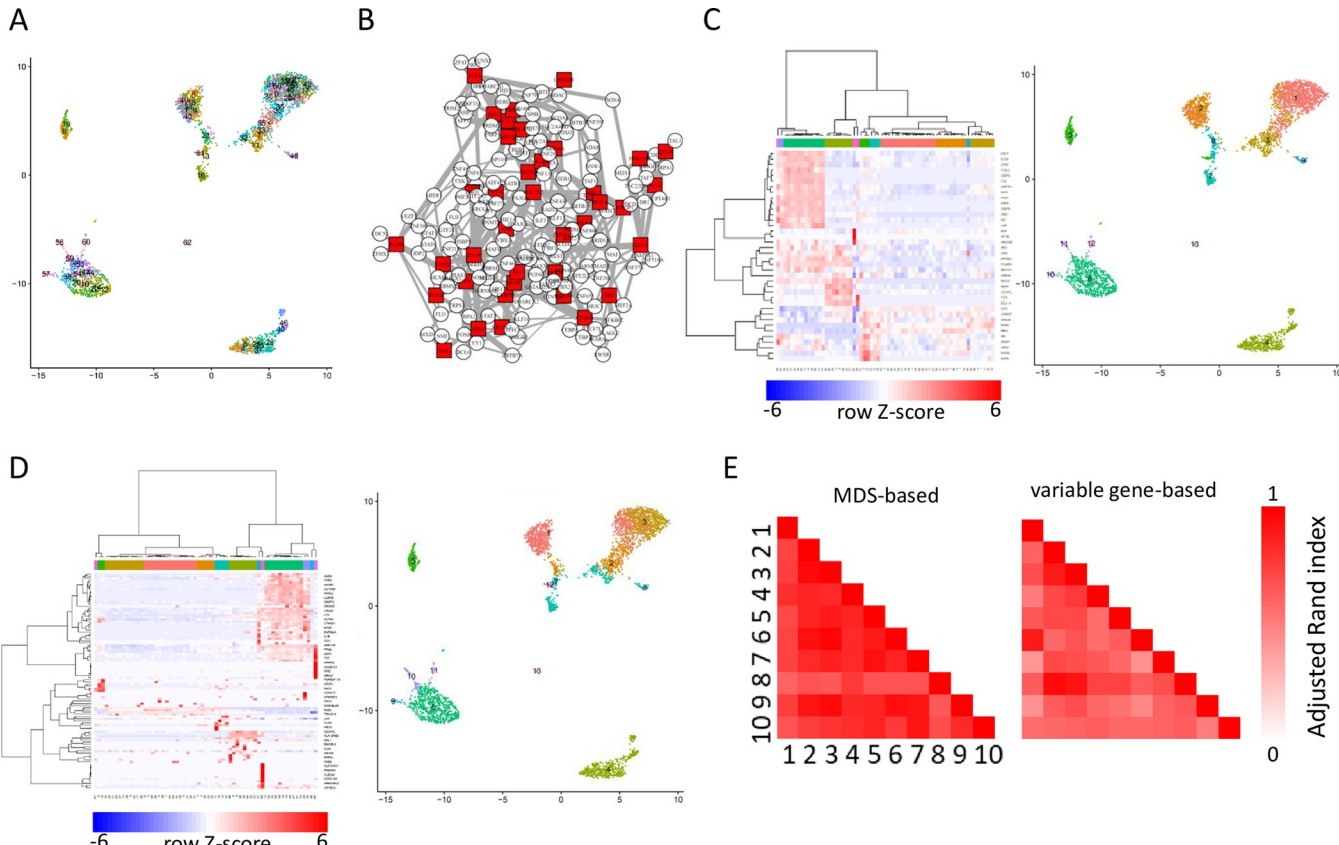

**Fig 4. BootCellNet robustly identifies cell types.** (A) Clumps of cells by using Louvain clustering with a resolution of 10. (B) GRN and its MDS (red squares) identified by BootCellNet. (C) An example of the heatmap representation of hierarchical clustering of cell clumps shown in panel A (left) and the resulting clusters (right) based on MDS. (D) An example of the heatmap of hierarchical clustering of cell clumps (left) and the resulting clusters (right) based on 100 genes with variable expression. (E) Adjusted Rand indexes calculated among 10 MDS-based clusterings (left) and 10 variable gene-based ones (right) were shown.

dendritic cells (DCs)[9]. This aligns with the cluster's association with myeloid DC signatures. Cluster 12, marked with PDGFRA and GATA1, is linked to a platelet signature. Cluster 13, identified by SPIB, TCF4, and IRF7 as markers, all of which are well-known TFs involved in plasmacytoid DC (pDC) development and function[10]. Consistently, the cluster is associated with pDC gene signature, indicating that the cluster represents pDC population.

To compare this result with reference-based supervised cell labeling, the cells were labeled by using SingleR with Human Cell Atlas dataset in celldex R package. The fine dataset contains 157 subtypes, resulting in 19 cell types as shown in Fig 5B. Most labels are consistent with the result from BootCellNet. However, the reference-based labeling misclassified certain populations, such as clusters of myeloid DCs (Cluster 10) and pDCs (Cluster 13) as monocytes and CD34-GMP/Pre-B cells, respectively. Moreover, it tended to subdivide cells into smaller populations. NK cells, represented by Cluster 5 in MDS-based clustering, were subdivided into several clusters with similar markers, such as killer-cell receptors (KLRs) and granzymes (GZMs). The activated CD8 T cell Cluster 7 in MDS-based clustering was mislabeled as a CD4 T cell. B cells, represented by Cluster 4 in MDS-based clustering, were also divided into several subpopulations, most with nearly identical markers. A population, B cell with memory phenotype, has unique markers IGHA1/2, implying class switching which is a characteristic of activated B cells. This implies that some populations might not be identified by MDS-based clustering.

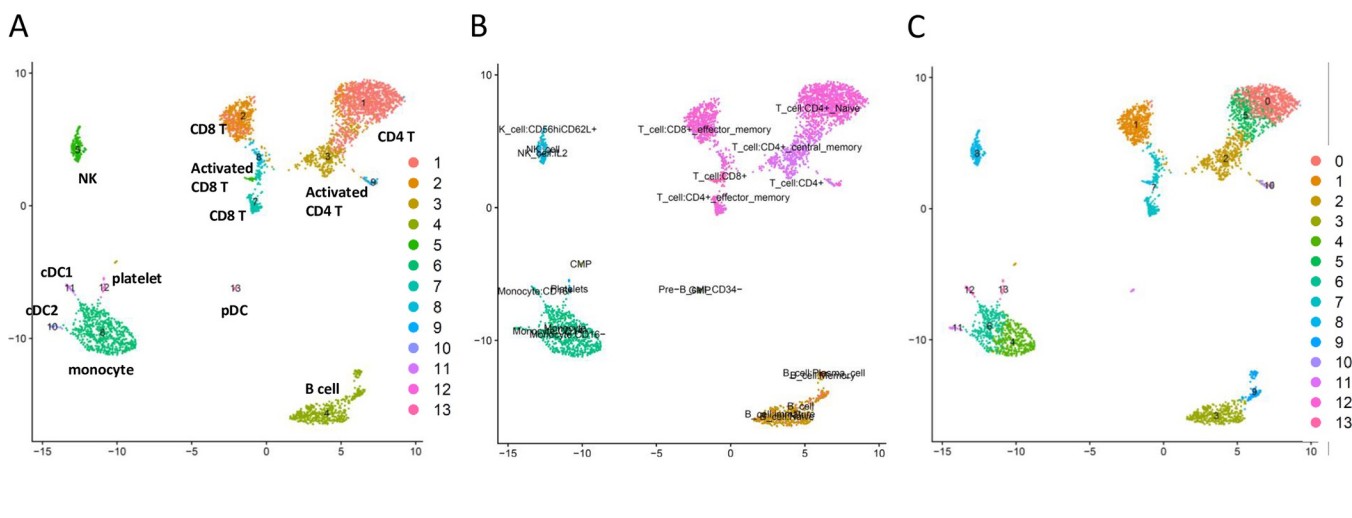

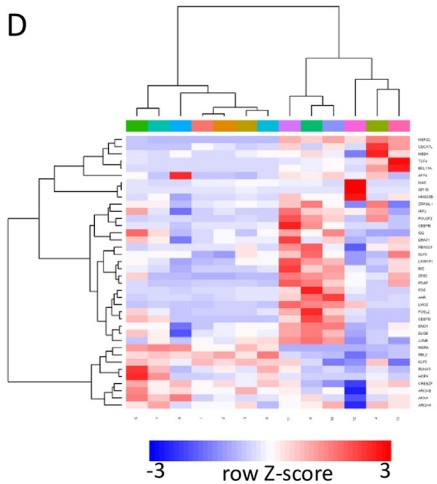

**Fig 5. Cell types in peripheral blood mononuclear cells identified by BootCellNet and its comparison to reference-based labeling and unsupervised clustering.** (A) The resulting clusters obtained by BootCellNet and their annotations described in the main text were shown. (B) Annotation of cells obtained by SingleR with the fine HumanCellAtlas dataset. (C) Clustering result obtained by Louvain clustering implemented in Seurat, with a resolution of 1.0. (D) Heatmap of average expression of MDS genes in each cluster obtained by BootCellNet. Color codes for columns are the same as those in panel A.

The result obtained by BootCellNet was also compared to that of unsupervised Louvain clustering (Fig 5C). We chose a resolution of 1.0 because it gave 14 clusters, a similar number to that of BootCellNet. Although the results were mostly similar, we found two notable differences. First, a cluster of pDC was missing. Those cells were mixed with cDC2 to form a cluster (Cluster 11 in Fig 5C). Second, Louvain clustering could distinguish memory B cells from naïve B cells (Clusters 9 and 3 in Fig 5C, respectively), while BootCellNet could not and mixed into one cluster (Cluster 4 in Fig 5A) as described above. These results imply that unsupervised clustering including Louvain clustering is effective enough to identify cell types, though there's no guiding principle to choose resolution and no rational explanation to separate or merge cells at the step of clustering. We will discuss these points later.

BootCellNet identifies MDS in GRN. This provides insights into how the differentiation and/or the function of cell types are regulated by checking the expression of genes in MDS as shown in Fig 5D. Comparing the B cell cluster 4 and the pDC cluster 13, TCF4, a TF important

for pDC development[10], was expressed more in Cluster 13. Cluster 12, representing platelets, has a high expression of GFI1B which regulates platelet function[11]. We found CEBPB, a TF that controls monocyte/macrophage differentiation, was higher in Clusters 6 and 11 of monocyte/macrophage compared to Cluster 10 of DCs. Id2 is expressed more in the NK cell cluster 5, compared to Clusters 1, 2, 3, 7, 8, and 9, all of which are clusters of T cells. Consistently, the TF is essential for NK cell maturation[12]. Cluster 7 also shows high Id2 expression, thought to a lesser extent. This is consistent with reports that Id2 is important for CD8+ memory T cell development[13] because the cluster represents an activated CD8 T cell population. Further, by assessing RUNX3 expression, we determined Cluster 2, 7, and 8, which show higher RUNX3 expression, to be CD8 T cells, whereas Cluster 1, 3, and 9 are identified as CD4 T cells, given that RUNX3 is important for CD8 T cell development[8]. By examining RORA expression in Cluster 1 and 3, we identified Cluster 3 as activated T cells, since RORA is notably expressed in Th17 population[14].

These findings collectively suggest that MDS-based clustering offers robust and accurate cell type identification, largely aligning with reference-based supervised labeling, while providing insights into the regulatory mechanisms of the cells through the identification of GRN and MDS.

## BootCellNet facilitates inference of developmental courses of hematopoiesis

The above result indicates that BootCellNet can identify cell types from scRNA-seq data containing multiple discrete cell types. To examine if BootCellNet can help to infer continuous spectra of cell states, we applied BootCellNet to scRNA-seq hematopoiesis data. The scRNA-seq data of myeloid cell development[15] was subjected to BootCellNet. The procedure resulted in a GRN of 110 TFs and MDS of 22 TFs (Fig 6A). The MDS includes TFs that are known for their importance in myeloid/erythroid development, such as Cebpe, Gata2, Gfi1b, Irf8, Jdp2, Klf1, Meis1, Myb, Sfpi1, and Sox4, further suggesting the robust inference GRN and its dominating TFs by BootCellNet.

The cells were clustered into 9 as shown in Fig 6B. By examining the expressed MDS TFs (Fig 6C), the marker genes (S4 Table), and associated GO terms (S5 Table), we identified Cluster 1, 4, 6, and 8 as stem cells, Cluster 2 as erythrocytes, Cluster 5 as megakaryocytes, Cluster 3 as monocytes, and Cluster 7 as neutrophils. Cluster 9 seems to be lymphocytes and dendritic cells, which are removed in the analyses performed in a report using the data[16].

Cluster 1 has Flt3 as a marker, as well as Sox4, Runx1, Cebpa, Lmo4, and Gata2, which are TFs known to be expressed in hematopoietic stem cells. Cluster 3, 4, and 7 are high in Sfpi1 expression, indicating that those clusters are myeloid lineages. However, the expression of Sox4 distinguishes Cluster 4 from others as more progenitor-like cells. Cluster 3 is associated with terms related to innate immune responses, and has Csf1r as a marker gene, as well as Irf8. Cluster 7 is associated with myeloid cell-related terms and has Fcnb and Elane genes as markers. The mentioned marker genes are characteristics of monocytic and neutrophilic cells, respectively[17], supporting the identification of Cluster 3 as monocytes and Cluster 7 as neutrophils. Comparing Cluster 3 and 7, the former is high in Irf8 expression, whereas the latter has higher expression of Jdp2 and Cebpe. This expression pattern of TFs is also a characteristic that distinguishes monocytes and neutrophils[17, 18]. The importance of higher expression of Cebpe in the neutrophil lineage has been suggested by using CellOracle, which is a cell lineage analysis method based on GRN and a mathematical dynamical system model[16]. The result demonstrates that BootCellNet efficiently infers MDS and their roles in development, without using mathematical dynamical system models.

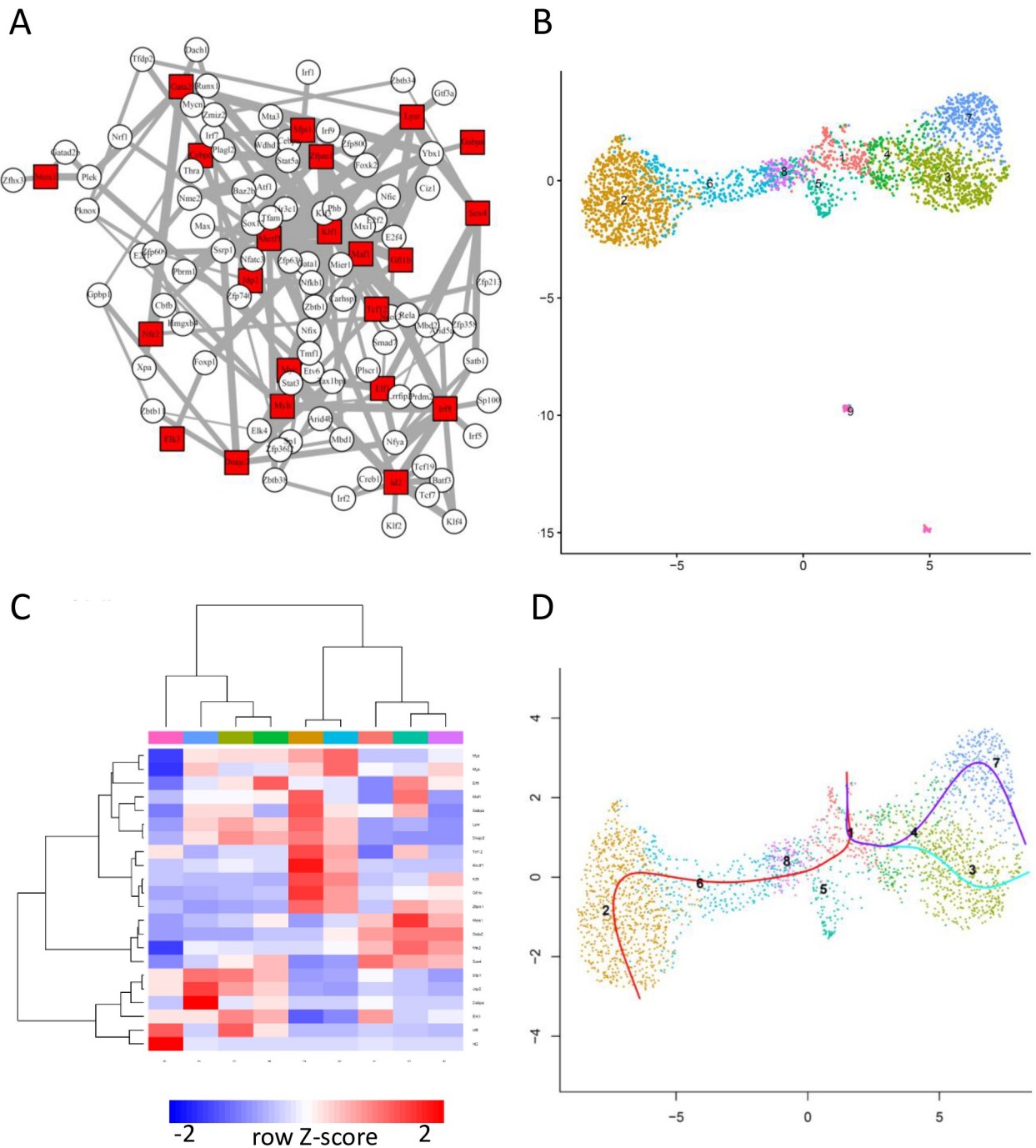

**Fig 6. BootCellNet facilitates the inference of the course of hematopoiesis.** (A) GRN and its MDS (red squares) identified by BootCellNet. (B) The resulting cell clusters. (C) Heatmap of average expression of MDS genes in each cluster obtained by BootCellNet. Color codes for columns are the same as those in panel B. (D) Lineage curves were obtained by Slingshot using the clusters by BootCellNet. One lineage curve, which spans from Cluster 1 to Cluster 9 is omitted, since Cluster 9 encompasses lymphoid cells.

We inferred the lineage trajectories by using the clusters identified by BootCellNet. Slingshot[19] was applied to the data, setting Cluster 1 as a starting point. This resulted in 4 lineage curves (Fig 6D; one lineage is omitted, see the Figure Legend for details). The result recapitulates known developmental trajectories and aligns with the analysis performed by using a

different method[16]. One trajectory represents the development of erythrocytes spanning from Cluster 1 to 2 through 8 and 6. The others represent that of monocytes and neutrophils spanning from Cluster 1, through 4, to Cluster 3 and 7, respectively. This result suggests that cell cluster identification by BootCellNet facilitates lineage trajectory inference.

## Automated cell type cluster identification by BootCellNet2

Two ambiguous points remain in the BootCellNet procedure. First, MDS is not necessarily uniquely determined. Second, we determined the number of clusters by visual inspection of the hierarchical clustering heatmap.

To circumvent the first problem, we use the idea of critical nodes and intermittent nodes. In the case where multiple MDSs exist, one can define critical nodes and intermittent nodes, which appear in all the MDSs or at least in one MDS[20]. A set of critical and intermittent nodes is uniquely determined. By using those sets, we could circumvent possible inconsistencies derived from the non-uniqueness of MDS.

To address the second point, we utilize multiscale bootstrapping to determine statistically significant clusters. The method provides approximately unbiased confidence for each clustering[21]. By providing a cutoff $p$-value in advance, an R package pvclust extracts statistically confident clusters of cell clumps. However, this leaves other clumps unclustered. Thus we compare the clusters given by pvclust with the clusters obtained by hierarchical clustering with varying numbers of clusters, and determine the number of clusters.

We implemented the above methods (Fig 7A), and the procedure was applied to the PBMC dataset. Using the same dataset as in Fig 2, we first checked whether the new procedure consistently gives a unique set of nodes. We counted the appearance of nodes, both critical and intermittent, in 10 specimens. As shown in Fig 7B, the number of nodes that consistently appeared 10 times was not significantly different from that of the former procedure. This indicates that the uniquely determined set of nodes is still influenced by noises and resampling in GRN reconstruction.

We then performed clustering of cell clumps using the critical and intermittent nodes instead of MDS. Also, we utilized multiscale bootstrap to determine the number of clusters. The R package pvclust gives clusters with relatively low approximately unbiased p-values. Based on the clustering, we determined the number of clusters as described in detail in Methods and shown briefly in Fig 7A. The obtained clusters were compared with each other and ARI was calculated. As shown in Fig 7B, the automated clustering procedure along with the calculation of critical and intermittent nodes (hereafter we call the nodes as control nodes) gave clustering results highly similar to each other. The numbers of clusters were stable, 16 or 17 except for one 14. This indicates that the procedure offered consistent clustering results. If we did not use the unique set of control nodes for clustering but instead used MDS, the number of clusters had more variations. ARIs were significantly higher in the combination of the automated clustering procedure and the use of the unique set of nodes than others (Fig 7C). This suggests the importance of the use of the unique set of nodes, although the set had variation as shown above. Taken together, these results indicate that the use of the automated clustering procedure together with the unique control node set offers more consistent clustering than the former procedure.

To evaluate the newly obtained clusters, we picked the same specimen as shown in Fig 5A. The resulting GRN with critical nodes and intermittent nodes is shown in Fig 8A. The GRN contains 186 TFs with 22 critical and 33 intermittent TFs. Using the nodes, cell clumps were clustered, obtaining 17 clusters. The clusters are mostly the same as those in Fig 5A. We checked the marker genes and associated GO terms (S6 and S7 Tables, respectively). Based on

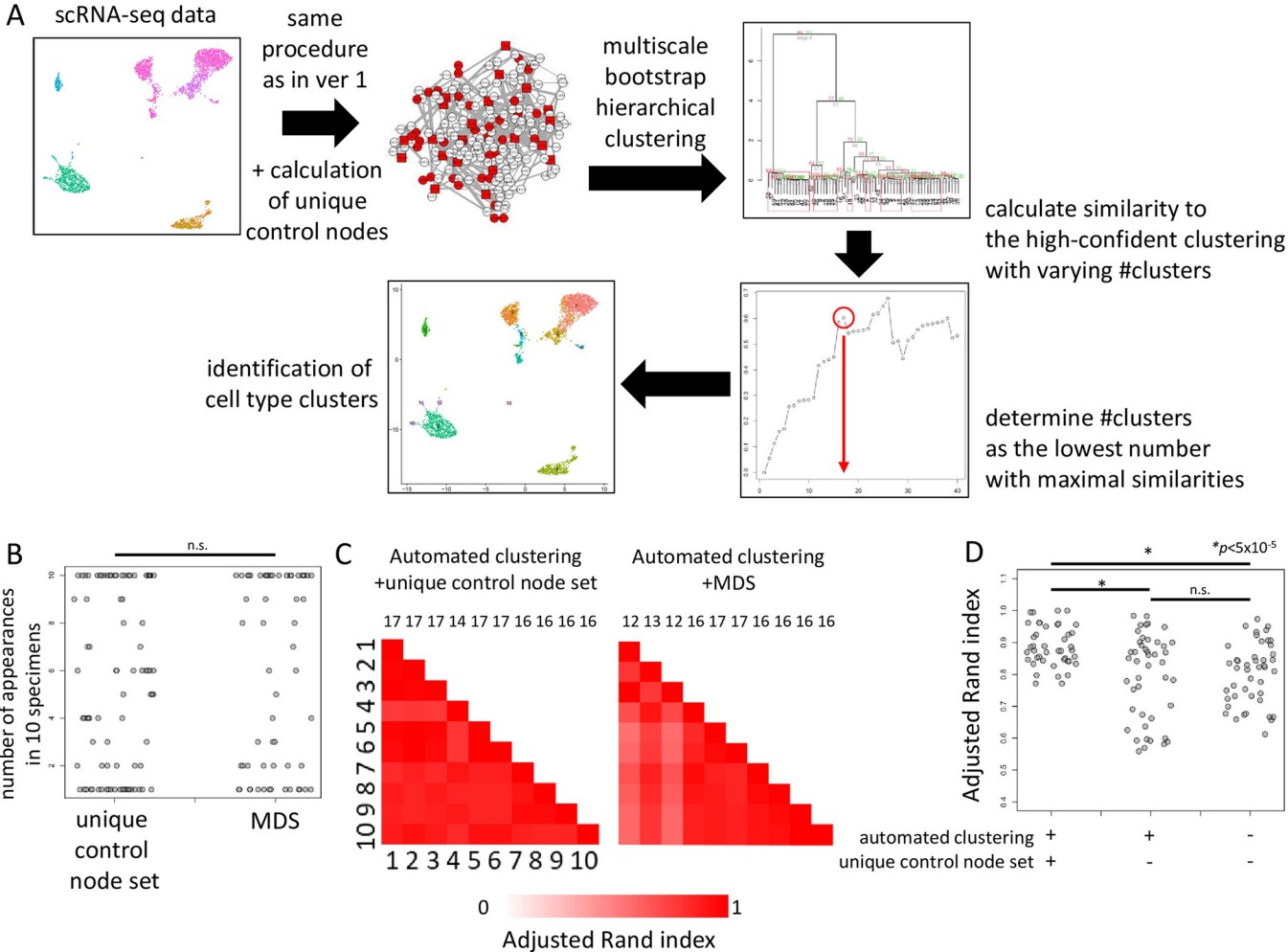

**Fig 7. BootCellNet2 and its validation.** (A) Outline of the procedure. Upto the GRN construction, the procedures are the same. From the constructed GRN, the unique set of control nodes consisting of critical nodes and intermittent nodes was computed. Cell clumps are obtained in the same manner as BootCellNet. Averaged expression levels of genes in the unique set are calculated and used for the clustering of cell clumps. Here the clustering was done with multiscale bootstrap to obtain highly confident clusters. The resulting high-confident clustering results are compared to clustering with varying numbers of clusters. The optimal number of clusters is determined as the lowest number of clusters with maximal similarities. (B) Number of appearances of genes in the unique set of control nodes or MDS in 10 specimens. No statistically significant difference was observed by Welch's two-sided *t*-test. (C) Adjusted Rand indexes calculated among 10 automated clusterings with (left) or without (right) using the unique set of control nodes are shown. The numbers above the heatmaps represent the number of clusters determined by the BootCellNet2 procedure. (D) Adjusted Rand indexes calculated among 10 automated clusterings with (left) or without (middle) using the unique set of control nodes, along with those without using either automated clustering and the unique set (right) are shown. *p < 5x10^-5 by Welch's two-sided *t*-test.

the marker genes and GO terms, the cell clusters were labeled as shown in Fig 8B. Clusters of T cells were more separated than in Fig 5A, and the novel clustering result enabled the identification of naïve, activated, central memory, and effector memory subsets, along with Cluster 1, 4, and 11, which represent naïve CD4 T cell subsets. As in the above analysis, we made use of markers as well as the set of nodes to explain why those clusters are separated (Fig 8C). Cluster 1 is high in JUNB which is critical for the survival of T cells[22], while Cluster 11 is high in NFE2L2/NRF2 which interfers activation of T cells[23] and Cluster 4 is not high for either, suggesting differential activation states of the three naïve T cell clusters. These results together suggest that the novel procedure BootCellNet2 offers automated and robust clustering and identification of biologically meaningful cell types.

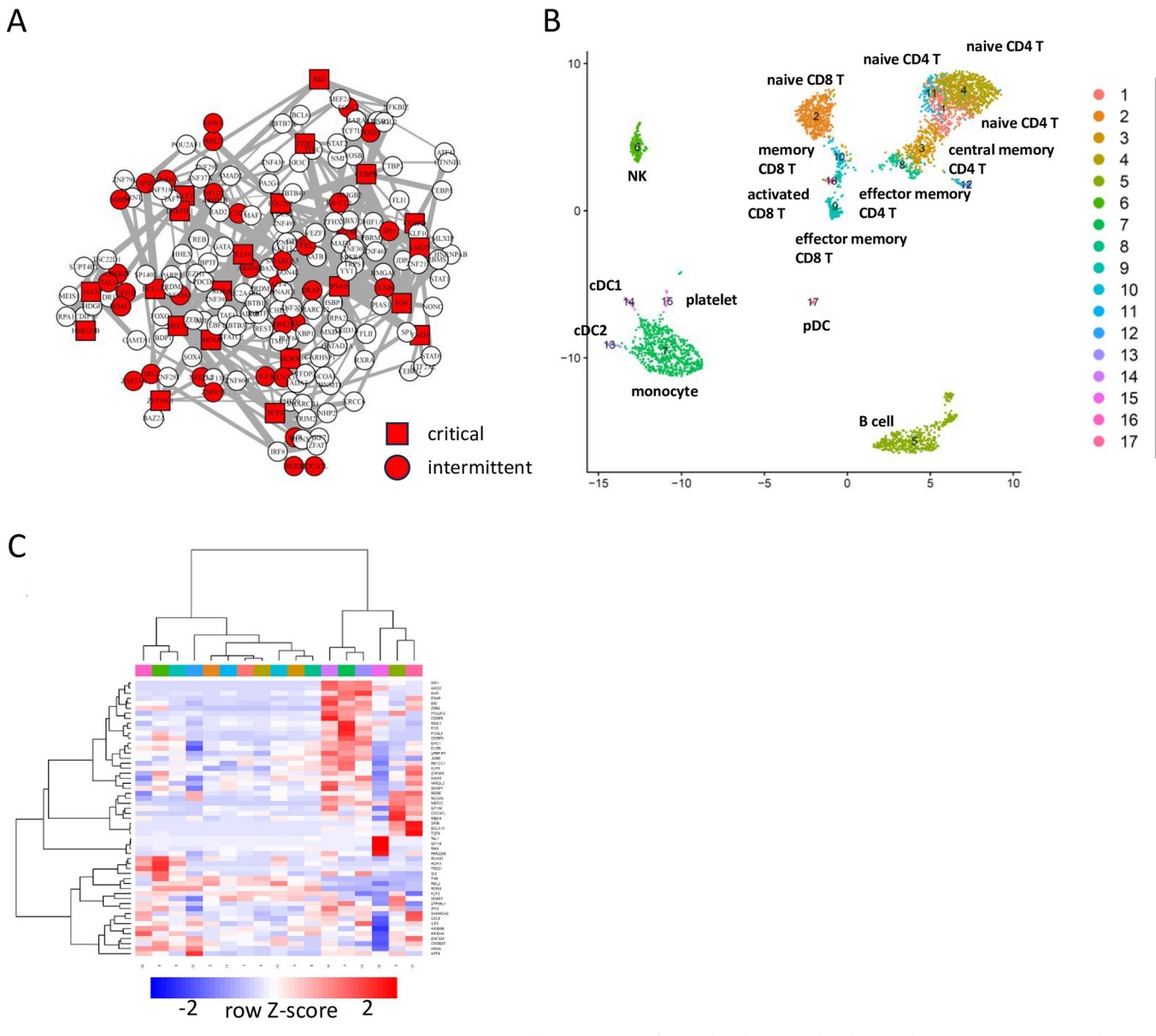

**Fig 8. Identification of cell type clusters by BootCellNet2.** (A) GRN and its unique set of control nodes (critical nodes as red squares, intermittent nodes as red circles) identified by BootCellNet2 are shown. (B) The resulting cell clusters. Annotations for each cluster described in the main text were shown. (C) Heatmap of average expression of genes in the unique set in each cluster obtained by BootCellNet2. Color codes for columns are the same as those in panel B.

## BootCellNet2 identified cell type clusters complying with CITE-seq only from transcriptome data

We further validated BootCellNet2 using CITE-seq data, which allows us to evaluate clustering results with antibody-derived tags. The BootCellNet2 procedure was applied to transcriptome data from the CITE-seq study reported[24]. The analysis identified a set of 29 control nodes, consisting of 19 critical nodes and 10 intermittent nodes (S1 Fig), and generated 10 clusters as shown in Fig 9A. We labeled the clusters based on their marker genes and associated GO terms for each cluster (S8 and S9 Tables, respectively), as well as CITE-seq results (Fig 9C). In comparison, we also performed clustering using the Louvain algorithm with a resolution of

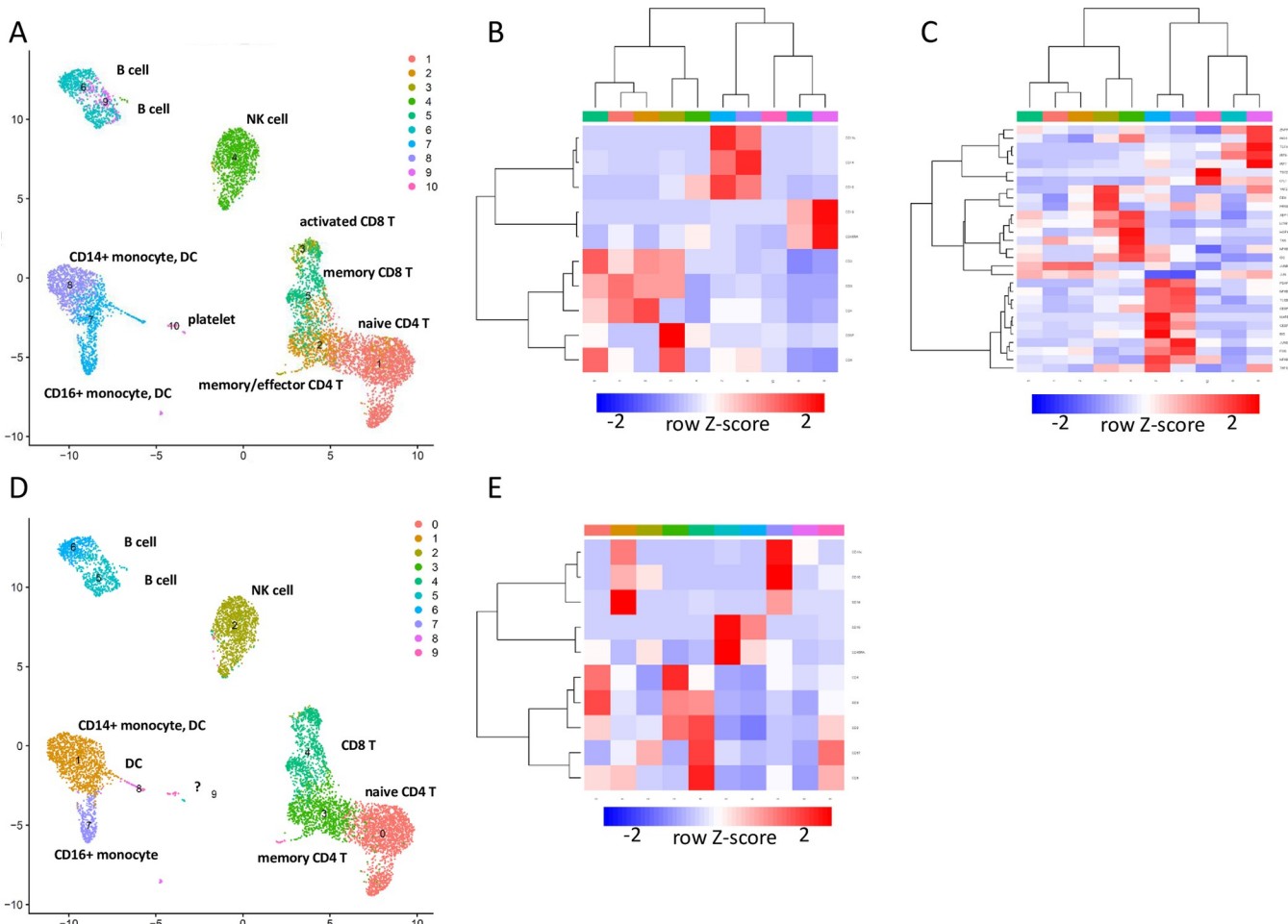

**Fig 9. BootCellNet2 gives clustering that comply more with CITE-seq results.** (A) The cell clusters obtained by BootCellNet2. Annotations for each cluster according to GO terms, marker genes, and antibody-derived tags were shown. (B) Heatmap of average expression of antibody-derived tags in each cluster obtained by BootCellNet2. Color codes for columns are the same as those in panel A. (C) Heatmap of average expression of the genes in the unique control set in each cluster obtained by BootCellNet2. Color codes for columns are the same as those in panel A. (D) The cell clusters obtained by the Louvain algorithm with a resolution of 0.7. Annotations for each cluster according to GO terms, marker genes, and antibody-derived tags were shown. (E) Heatmap of average expression of antibody-derived tags in each cluster obtained by the Louvain algorithm. Color codes for columns are the same as those in panel D.

0.7, which resulted in 10 clusters, the same number of clusters as BootCellNet2. The clusters were labeled as shown in Fig 9D, according to the marker genes, associated GO terms, and CITE-seq results (S10 and S11 Tables and Fig 9E, respectively). The results suggest that the two methods offered largely comparable results.

A few key differences, however, were found. BootCellNet2 identified distinct states of T cells similar to those observed in another PBMC dataset analysis shown above (see Fig 8B), which was not the case with Louvain clustering. Moreover, CITE-seq results suggest clear separation in CD4 and CD8 expressions within the T cell clusters identified by BootCellNet2 (Cluster 1, 2, 3, and 5, as shown in Fig 9B); Clusters 3 and 5 exhibited low CD4 and high CD8, whereas Clusters 1 and 2 showed high CD4 and low CD8 expression. The difference between the states of CD4 and CD8 T cells was also evident from CD45RA expression known to be high in naïve T cells (high in Cluster 1 and low in Cluster 2) and CD57 expression known to be high in activated T cells (high in Cluster 3 and low in Cluster 5), respectively. Expression of HMGB2, which is highly expressed in effector and memory T cells compared to naïve T cells

[25], further supports these findings (Fig 9C). Additionally, platelets were identified by Boot-CellNet2 but not by Louvain. These results collectively imply that BootCellNet2 clustering achieves more accurate cell type delineation, aligning more closely with antibody-derived tag results than unsupervised clustering.

## Identification of characteristic cell types in COVID-19 patients by BootCellNet2

We applied BootCellNet2 to scRNA-seq data of bronchoalveolar lavage fluid cells from COVID-19 patients[26] to test whether BootCellNet2 can identify cell types in scRNA-seq data where discrete and continuous cell types/states are mixed. The cells were clustered into 32 in the original article. BootCellNet2 gave the GRN of 161 TFs with 21 critical nodes and 13 intermittent nodes (Fig 10A). The automated clustering procedure offered 17 clusters of the cell clumps (Fig 10B). Marker genes were identified for each cluster, and the GO terms associated with the markers of each cluster were checked (S12 and S13 Tables, respectively).

Clusters 1, 2, 3, 5, 8, 13, 15, and 16 were associated with myeloid cell-related signatures. Compared to the clusters in the original report (reproduced in S2 Fig, encircled as "macro-phage/monocyte", "neutrophil", and "DC"), the clustering by BootCellNet2 has a smaller number of clusters while identifying myeloid cells with different characteristics.

Clusters 1, 3, and 8 were associated with inflammatory immune responses and/or characteristics of macrophage/monocyte/DC thus identified as myeloid cells. Clusters 1, 3, and 8 have high expression of FOS, IRF1, and NFKB2, which are TFs involved in inflammatory immune response (Fig 10C). The clusters are significantly enriched with cells from severe patients (Fig 10D). The associated terms are also related to inflammatory responses. In contrast, Clusters 2, 5, 13, 15, and 16 are enriched with cells from mild and/or healthy individuals. Clusters 5, 6, 13, 14, and 16 have HLA genes as markers and are associated with DC/monocyte signatures, implying the clusters contain antigen-presenting myeloid cells. Clusters 5, 13, and 16 have either MARCO, CEBPA, and PPARG as markers, indicating that the clusters are alveolar macrophages since PPARG is an essential TF for alveolar macrophage development[27]. Clusters 5, 6, 13, 14, and 16 did not show high expression of inflammatory TFs such as FOS, IRF1, and NFKB2, in contrast to Clusters 1, 4, and 8. This implies that the myeloid cells in healthy individuals or patients with mild symptoms are less inflammatory than those in individuals with severe disease, consistent with former reports showing that inflammation and cytokine storms are the cause of severe COVID-19[28].

Cluster 2 is associated with myeloid and dendritic cell signatures. It has CD19 as a marker. Indeed, the cluster may contain pDC, myeloid DC, and B cells, which were clustered separately in the original report (S2 Fig), suggesting a probable failure of the BootCellNet2 procedure in the identification of cell types. Clusters 4 and 15 are associated with T cell signature and have genes for components of T cell receptors such as CD3D/E/G, LCK, LAT, and ZAP70 as markers. They also have CD8A/B and RUNX3 as markers, indicating that those clusters are CD8 T cell populations. Further, all of them have genes for granzymes and IFN-g as markers, suggesting that the clusters represent activated CD8 T cells.

To examine the differences among those activated T cells clustered separately, we compared Cluster 4 to Cluster 15 and found that the latter, which is enriched with cells from patients with mild symptoms, but not the former, which is enriched with cells from patients with severe symptoms, has ID2, TCF7, and CCR5 as markers, which are highly expressed in effector memory CD8 T cells[13, 29, 30]. We also examined the expression of IL7R (encoding IL-7Ralpha) and PDCD1 (encoding PD-1) and found that Cluster 15 has both as markers while Cluster 4 has only PDCD1, indicating that Cluster 4 is exhausted T cells [31]. Notably, examination of

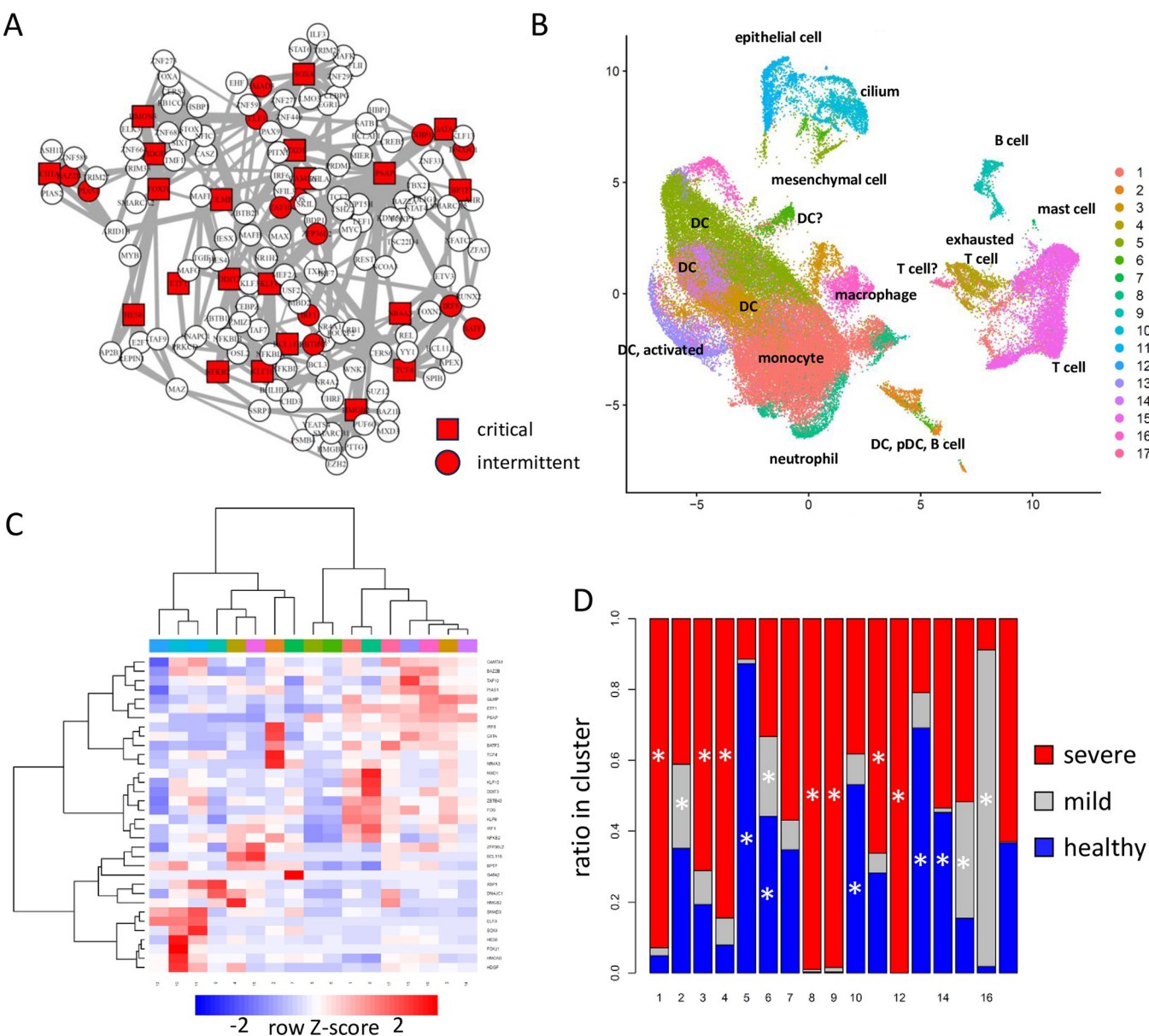

**Fig 10. Identification of cell types in bronchoalveolar lavage fluid scRNA-seq data by BootCellNet2.** (A) GRN and its MDS (red squares) identified by BootCellNet. (B) The resulting cell clusters. (C) Heatmap of average expression of MDS genes in each cluster obtained by BootCellNet. Color codes for columns are the same as those in panel B. (D) Distribution of cells from each class of patients in each cluster. Statistical significances of enrichment of cells from each origin were calculated by Fisher's exact test with Bonferroni correction. Asterisks were shown when p<5x10^-4.

the expression of the control nodes in the GRN also showed that HMGB2, which is indispensable for the development of exhausted T cells[25], is higher in Cluster 4 than in Cluster 15 (Fig 10C). These results collectively suggest that induction of effector memory CD8 T cell upon SARS-CoV-2 infection is critical for controlling disease to mild symptoms, and an increase of exhausted T cells is detrimental for disease control, consistent with former reports[26, 32, 33].

Taken together, the results indicate that BootCellNet2 successfully identifies clusters of cells specific to patients with different symptoms, and offers explanations of the difference in symptoms that are consistent with preceding reports.

## Discussion

In the current study, a method named BootCellNet has been developed, which combines smoothing, resampling, GRN inference, computation of a set of genes controlling the entire network, and clustering. We demonstrated the robustness of our method, particularly given the probabilistic steps involved and the presence of noise within the data. We also performed an expert-based evaluation of the clusters by manually labeling the clusters. This ensured that despite the probabilistic nature of our method, the clusters identified are not only statistically robust but also biologically meaningful.

Since the method consists of multiple steps, the choice of algorithms will affect the results of the procedure. We used $k$-NN smoothing to reduce the noise in scRNA-seq. We performed the smoothing on the 2-dimensional UMAP plane, but one can do the same in a higher dimension[34]. One can also use different smoothing methods, such as $k$-NN graph-based MetaCell algorithm[35], or data imputation methods such as MAGIC[36] and DeepImpute[37].

In BootCellNet, GRN reconstruction is performed with the ARACNe method, because of its low computational load. As in the current study, NestBoot utilizes a nested bootstrap to control FDR in GRN inference[38], and they showed that the bootstrapping procedure improved the accuracy of the GRN inference by various inference methods such as GENIE3 [39]. Thus, the incorporation of methods other than ARACNe in BootCellNet may improve the GRN inference accuracy.

The use of MDS as a set of genes that controls the dynamics of the whole network needs some caution. The controllability of a network by its MDS requires a premise that each MDS node can control its adjacent nodes individually[5]. In the case where nodes are TFs, this may be justified by assuming that a TF controls each target gene through individually controlled accessibility, although the assumption should be verified independently.

Nevertheless, we demonstrated that the utilization of expression levels of only several tens of MDS genes or the unique set of control nodes was sufficient for robust clustering of cells while using one hundred genes with variable expression was not. This implies that MDS can provide a dimensionality reduction of scRNA-seq data.

We inferred the network of TF genes in the current study, because we used transcriptome data, which is principally controlled by TFs. But this could be extended by incorporating not only TFs, but also co-factors, post-transcriptional regulators, and soluble mediators that control TF expression and/or activity. The incorporation of these factors also conforms to the picture where transcriptome activity is controlled principally on the mRNA level. Indeed, some post-transcriptional regulators, such as ZFP36L1 and ZFP36L2 were included and found to be a part of MDS. The incorporation of more factors will help to uncover dynamic picture of the regulatory network.

Despite the room for improvement and caution, and although the method is relatively simpler than other sophisticated methods, we demonstrated that BootCellNet effectively worked for a variety of datasets with different characteristics. Notably, the proposed method excels in explainability, not only clustering cells but also explaining why they are clustered based on the GRN, and providing characterization of what type of population they represent. Along with automating the determination of the number of clusters, the method resolves the common problem in unsupervised clustering which does not provide an answer for the appropriate number of clusters and/or an explanation of why some cells are clustered together but others are separated. For example, the clusters can be explained by factors such as GFI1B, CEBPB, and RUNX3 in clustering PBMC data, by Cebpe in clustering hematopoiesis data, and by HMGB2 in clustering COVID-19 data, allowing the identification of what type of cells the clusters represent and what characteristics they possess, even within similar cell types, without

looking at GO terms, using information from GRN/MDS. This demonstrates that BootCellNet can be a useful complementary for other methods of clustering, supervised labeling, and lineage inference.

Also, BootCellNet, especially BootCellNet2, is suitable for automation. The method will be utilized for automated and routine annotation of large-scale scRNA-seq cell data. Moreover, BootCellNet addresses an increasing need for explainable automated data processing[40], since it has a firm basis in statistics and network dynamics. Although there is a strict need for manual inspection and experimental verification[1], the proposed method will accelerate the generation of the working hypotheses from a large amount of scRNA-seq data.

## Materials and Methods

### Outline of the BootCellNet procedure

The proposed procedure is outlined in Fig 1. The procedure, BootCellNet, utilizes smoothing of scRNA-seq expression data and resampling to reduce noise in the dataset and to facilitate a robust inference of gene regulatory network (GRN), in combination with computation of minimum dominating set (MDS). Using the MDS, BootCellNet performs clustering of cells to identify cell types and states.

BootCellNet2 is outlined in Fig 7A. The procedure utilizes a unique set of control nodes consisting of critical nodes and intermittent nodes, instead of MDS, and the clustering is automated as described below.

### Estimation of statistical significance by using resampling

The nature of the BootCellNet procedure enables statistical evaluation of inferred GRNs. BootCellNet repeatedly samples cells from the smoothed scRNA-seq dataset and performs GRN inference. Some nodes (genes) and edges (regulatory interactions) frequently show up in each resampled set, while others do only occasionally. At the same time, we shuffle gene names and perform the same inferences repeatedly to evaluate the occurrence of false positives.

We first performed the $k$-NN smoothing on 2-dimensional UMAP space. The $k$ is set to a value small enough to find a cell population having about $k$ cells in a dataset. In this regard, the $k$ determines the resolution of the analysis. For given $k$ and 2-dimensional UMAP space, for each cell, $k$-nearest neighbors in the UMAP space with Euclidean distance are selected, and the average of the expression levels is computed to obtain $k$-NN smoothed cells. The decimal part of the averaged expression levels was truncated to obtain expression levels in integers.

Next, we resample $N$ $k$-NN smoothed cells for $M$ times to infer GRN or for $\hat{M}$ times to calculate the support values of appearances of genes/nodes and regulations/edges. In the current study, the resampling number $N$ was set to from one-third to half of the total number of cells in the dataset, but no more than 20000 to reduce the computation load. For each resampled set, we compute variable expression genes and then infer GRN. We also perform shuffling of gene names in each resampled set to calculate the support.

Let the resampled sets be $X_m = \{x_{m1}, x_{m2}, \ldots, x_{mN}\}$, where $m = 1, 2, \ldots, \hat{M}$, and $x$ be a vector of $k$-NN smoothed gene expressions of a cell. Shuffling of gene names in $X_m$ for $K$ times is performed to generate nested resampling sets $X_{mk}$ ($k = 1,2,\ldots K$) [41]. Then we obtain $S(X_{mk}; \theta)$, which takes 1 if a hypothesis $\theta$ is supported by the set $X_{mk}$ or 0 if not. In the current case, $\theta$ is a hypothesis that we find a gene as a variable expression gene or one that we find a regulatory relationship (i.e. an edge of GRN) in a GRN inferred from $X_{mk}$. The parameter to be estimated is $\lambda(\theta) = \frac{1}{MK} \sum_{m,k} S(X_{mk}; \theta)$, representing how often we occasionally have $\theta$. Once we estimate the parameter, we can compute a $p$-value by supposing the event obeys a Poisson distribution

with parameter $\lambda M$. However, the rate of occasional appearances of genes is equal to (the number of variable expression genes selected) / (the number of total genes), and thus there's no need to compute by resampling. Hence, we computed the $\lambda$ only for regulations/edges in GRNs. The obtained $p$-values were adjusted by the Benjamini-Yekutieli method to obtain $q$-values. Nodes and edges with $q<1e-05$ were considered significant, and non-significant nodes and edges were removed.

In the implementation of this paper, resampling of cells was performed using a function named BootCellNet, and shuffling of gene names was carried out in a function named AggregateBCN. For inferring GRN, resampling was done with $M = 100$, but the shuffling of gene names was not performed ($K = 0$). In the estimation of the support, resampling was performed with $\hat{M} = 1000$, and gene name shuffling was carried out with $K = 100$, obtaining $10^5$ resampled specimens.

## Inference of gene regulatory network

For each resampled specimen, 2000 genes with variable expression levels were selected by using FindVariableFeature function in Seurat[42] with vst method. From the set of genes, we only focused on transcription factors (TFs). The list of TFs was retrieved from a reported list of human TFs[43] in the case of human dataset analysis, and from AnimalTFDB 4.0[44] in the case of mouse dataset analysis. The TFs with variable expression levels were subjected to GRN inference.

GRN is inferred by using ARACNe method[45] implemented in an R package minet[46] with the estimation of mutual information by Miller-Madow asymptotic bias corrected empirical estimator and an equal width discretization, as recommended in a report[47].

## Computation of minimal dominating set (MDS)

The minimum dominating set (MDS) is a graph theoretical concept, which is the smallest subset of vertices in a graph such that every other vertex is adjacent to at least one vertex in this subset. MDS can be computed exactly by using an integer programming formulation described elsewhere[48]. We implemented the method using the R package lpSolve. Although the problem is known as an NP-hard problem, the number of genes in the GRN is typically less than 200 and thus MDS is computable in a laptop computer. We also implemented the computation of critical and intermittent nodes. To obtain the nodes, the computation of MDS is repeated for the defined times (15 times as default, by setting num.bin.solns = 15 in the function lp of the lpSolve package), and nodes identified as MDS in all the computations are defined as critical nodes, and those in at least one computation are defined as intermittent nodes.

## Clustering of cell clumps

In the BootCellNet procedure, we start from small clusters and aggregate them. The starting small clusters (here we call them clumps to distinguish them from the final clusters) were computed by using Louvain algorithm with a high resolution implemented in Seurat. The resolution was determined so that the size of the smallest clump was not much lower than $k$ (we use 80% of $k$ as a default procedure), which is the parameter used in the smoothing.

After obtaining the cell clumps, expression levels of MDS genes were averaged over each cluster. Using the expression levels, the clumps are hierarchically clustered by using Ward's algorithm with a value of Pearson correlation subtracted from 1 as a distance. The number of clusters of clumps was determined by visual inspection.

In the BootCellNet2 procedure, an R package pvclust is used to compute approximately unbiased p values for clusters of clumps with the multiscale bootstrap algorithm. A function

pvpick in the pvclust package extracts clusters having p values less than a given cutoff to offer a high-confident clustering. ARIs are computed between the high-confident clustering result given by pvclust/pvpick and those given by a function rect.hclust with varying numbers of clusters. Then the number of clusters of clumps is determined as the lowest number with a maximal ARI (see Fig 7A).

### Single-cell RNA-seq data processing

The peripheral blood mononuclear cell (PBMC) dataset was provided by 10x Genomics (https://cf.10xgenomics.com/samples/cell-exp/3.1.0/manual_5k_pbmc_NGSC3_ch5/manual_5k_pbmc_NGSC3_ch5_filtered_feature_bc_matrix.tar.gz). Expression levels and metadata were aggregated and processed with a standard workflow by using Seurat. The processed data containing 3707 cells was then subjected to BootCellNet with $k = 10$ and $N = 3000$. The starting 63 cell clumps were given with a resolution of 10, which was determined according to the criterion described above.

The myeloid/erythroid development scRNA-seq dataset[15] was retrieved from a repository (Paul_Cell_MARSseq_GSE72857.RData, https://github.com/theislab/scAnalysisTutorial). The data, comprised of 2730 cells, were processed with a standard workflow by using Seurat. The processed data were subjected to BootCellNet procedure with $k = 10$ and $N = 2000$. The starting 62 cell clumps were obtained with a resolution of 10.

The CITE-seq data of PBMC[24] was retrieved from NCBI GEO under accession number GSE100866. The data is comprised of scRNA-seq data and antibody-derived tag sequencing data. These data were aggregated and processed with a standard workflow by using Seurat. The processed data containing 7690 cells after filtering was then subjected to BootCellNet with $k = 10$ and $N = 5000$. The starting 66 cell clumps were obtained by using Louvain clustering with a resolution of 9.

The single-cell RNA-seq data of COVID-19 patients[26] were obtained from UCSC Cell Browser (http://cells.ucsc.edu/covid19-balf/nCoV.rds). The data is comprised of 66452 cells in total of bronchoalveolar lavage fluid from 4 healthy individuals, 3 mild and 6 severe patients. The data were analyzed by using Seurat and were subjected to BootCellNet with $k = 20$ and $N = 20000$. The starting 113 cell clumps were given by using Louvain clustering with a resolution of 10.

As a comparison, supervised cell type annotation was performed by using SingleR[49] with Human Primary Cell Atlas fine subtype labels in an R package celldex[49].

### Transcription factor binding site (TFBS), Gene Ontology (GO) enrichment and lineage analyses

Marker genes were obtained by using FindAllMarkers function in Seurat. We used Homer[50] for TFBS and GO analysis of the marker genes of each cluster. Promoters are defined as from -500 to +200 of transcription start sites. For lineage analysis, Slingshot[19] was utilized with default parameters.

### Statistical comparison of clustering results

Adjusted Rand index[51] was calculated to compare two clustering results.

### Supporting information

**S1 Fig. GRN reconstructed from CITE-seq data and its unique set of control nodes (critical nodes as red squares, intermittent nodes as red circles) identified by BootCellNet2 are**

**shown.**
(TIF)

**S2 Fig. The original clustering result of BAL scRNA-seq in Liao et al.** The result was reproduced from the deposited data and is shown (left). Cell type labeling based on the original paper is shown. As a comparison, the MDS-based clustering (same as Fig 9B) is also shown.
(TIF)

**S1 Table. Marker genes of each MDS-based cluster of PBMCs.** The output of FindAllMarkers function in Seurat was printed as a table.
(XLSX)

**S2 Table. Gene Ontology terms for marker genes of each MDS-based cluster of PBMCs.** Each sheet corresponds to each cluster.
(XLSX)

**S3 Table. TFBSs enriched in promters of marker genes of each MDS-based cluster of PBMCs.** Each sheet corresponds to each cluster.
(XLSX)

**S4 Table. Marker genes of each MDS-based cluster of cells undergoing hematopoiesis.** The output of FindAllMarkers function in Seurat was printed as a table.
(XLSX)

**S5 Table. Gene Ontology terms for marker genes of each MDS-based cluster cluster of cells undergoing hematopoiesis.** Each sheet corresponds to each cluster.
(XLSX)

**S6 Table. Marker genes of each BootCellNet2 clustering results of PBMCs.** The output of FindAllMarkers function in Seurat was printed as a table.
(XLSX)

**S7 Table. Gene Ontology terms for marker genes of PBMCs.** Each sheet corresponds to each cluster.
(XLSX)

**S8 Table. Marker genes of each BootCellNet2 clustering results of PBMCs based on CITE-seq.** The output of FindAllMarkers function in Seurat was printed as a table.
(XLSX)

**S9 Table. Gene Ontology terms for marker genes as in S8 Table.** Each sheet corresponds to each cluster.
(XLSX)

**S10 Table. Marker genes of each Louvain clustering results of PBMCs based on CITE-seq.** The output of FindAllMarkers function in Seurat was printed as a table.
(XLSX)

**S11 Table. Gene Ontology terms for marker genes as in S10 Table.** Each sheet corresponds to each cluster.
(XLSX)

**S12 Table. Marker genes of each BootCellNet2 cluster of BAL cells from COVID-19 patients.** The output of FindAllMarkers function in Seurat was printed as a table.
(XLSX)

**S13 Table. Gene Ontology terms for marker genes of BAL cells from COVID-19 patients.**
Each sheet corresponds to each cluster.
(XLSX)

## Acknowledgments

The authors thank Drs. Yasuyuki Kida, Nobuhito Mori, and Kentaro Kawata for discussions, Mr. Akihiko Fujii for technical assistance, and Dr. Yasuko Ozaki for secretarial assistance. Computational time was in part provided by the NIG supercomputer at ROIS National Institute of Genetics and by Human Genome Center, the Institute of Medical Science, the University of Tokyo.

## Author Contributions

**Conceptualization:** Yutaro Kumagai.

**Data curation:** Yutaro Kumagai.

**Funding acquisition:** Yutaro Kumagai.

**Investigation:** Yutaro Kumagai.

**Methodology:** Yutaro Kumagai.

**Software:** Yutaro Kumagai.

**Validation:** Yutaro Kumagai.

**Visualization:** Yutaro Kumagai.

**Writing – original draft:** Yutaro Kumagai.

**Writing – review & editing:** Yutaro Kumagai.

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
