## [Decision Letter · Decision Letter 0]

28 Mar 2024

Dear Dr Kumagai,

Thank you very much for submitting your manuscript "BootCellNet, a resampling-based procedure, promotes unsupervised identification of cell populations via robust inference of gene regulatory networks." for consideration at PLOS Computational Biology.

As with all papers reviewed by the journal, your manuscript was reviewed by members of the editorial board and by several independent reviewers. In light of the reviews (below this email), we would like to invite the resubmission of a significantly-revised version that takes into account the reviewers' comments.

In particular, as requested by Reviewer 2, pay particular attention to the reproducibility of your analysis, to the sharing of all codes and data required in this respect, and to make the corresponding tutorial more easy to follow.

We cannot make any decision about publication until we have seen the revised manuscript and your response to the reviewers' comments. Your revised manuscript will be sent to reviewers for further evaluation.

Sincerely,

Denis Thieffry, PhD

Academic Editor

PLOS Computational Biology

Pedro Mendes

Section Editor

PLOS Computational Biology

Reviewer's Responses to Questions

**Comments to the Authors:**

Reviewer #1: The work presents a novel method for inference of gene regulatory networks and cell types from single-cell RNA sequencing. The paper relies, among other things, on unsupervised machine-learning techniques such as clustering. However, it solves two major challenges with that kind of approach: inferring the number of clusters and reporting interpretable results.

The authors compare their unsupervised methods with a supervised procedure in Figure 5. However, they mention that there are also unsupervised methods in the literature. They should compare their approach with these, too. As a suggestion, they could contrast their results in the section on hematopoiesis and COVID-19 with the results found by other methods. This would make those sections more pleasant to read. The authors can skip this suggestion, but they should compare their method with some unsupervised ones at some point.

In the introduction, the authors state, "However, these methods do not provide an answer for the appropriate number of clusters and/or an explanation of why some cells are clustered together, but others are separated, leaving uncertainty in the analysis." That point was made throughout the results. However, it would be interesting to reiterate that point in the discussion with a sentence or two.

The authors could give a longer explanation of the k-NN. What are the features, and how is the smoothing done?

Minor comments: In the paragraph of lines 149-160, figure references are all shifted. 4B=4C, 4C=4D, 4D=4E.

Figure 3C right is missing an x-label.

Since I cite the sentence in lines 63-65, avoid using "and/or." There is a coma before "but."

Suggestions: Remove the word "very" from line 34. This is a very weak word (irony intended) that contributes nothing to the sentence.

I'm also sending this review as an attachment.

Reviewer #2: The author proposes a procedure/method/pipeline composed of multiple steps for the identification and inference of gene regulatory networks, from scRNA-seq.

The topic is of relevance since there's a always need for better methods to automate the clustering and analysis of data.

However, the document could be better written, providing a contextualization during each step of the pipeline.

Also, the background could better motivate the need for the tool (with appropriate references) and a better view of the existing tools, justifying the need for another tool.

One example of such is the phrase "There are several methods to explore pivotal elements..." which lacks a reference (if possible a review), especially if the methods are not being mentioned in this document.

Generically, it is a challenging exercise for the reader to understand if this document presents an analysis on specific datasets, or a generic software capable of being applied directly to other data by the wider community.

First and foremost, this should be clarified.

Phrases such as "Current clustering methods still suffer from uncertainty in determining the appropriate number of clusters" should be avoided, since the proposed method does not address this point also.

This problem is typically addressed by statistical methods, such Silhouette, Elbow, Sum of squares, etc.

The particular method used is not specified.

In page 7 it is mentioned that "By visual inspection of the heatmap of the MDS expression, we determine the number of clusters of clumps as 13". Visual inspection should not be the adequate criteria for a software/pipeline.

Regarding a proper comparison with existing software, it is only performed against SingleR, highlighting a particular case where the proposed method better captures the number of cell populations.

There are many software tools for clustering, GRN inference, and cell type identification using scRNA-seq data.

Finally, software-wise, this is a tool/method manuscript. So I really tried to test it, with the following difficulties.

- I managed to install R (despite being 2024).

- I also managed to install all the dependencies/packages to be able to run BoolCellNet, despite:

. dependency management in R is non-existent;

. some of the dependencies being discontinued, forcing me to find the correct old version .tar.gz and installing it manually.

But when finally running it, I got a file not found error since, I guess, some files (and the necessary directory structure) are missing from the github repo.

The software is currently not able to reproduce the results presented in the document, nor is it at a stage to be generically used by the wider community.

CONCLUSION:

The method does seem promising, but I do not recommend the acceptance of this document as is, without first do: 1) an overall improvement of the software and its tutorial; 2) a better placement/comparison of the software highlighting its advantage w.r.t. existing ones.

Reviewer #3: The author presents a pipeline to identify cell types from scRNA-seq data by combining several methods for smoothing, sampling, GRN inference and clustering. The main idea is to increase the robustness of GRN inference by smoothing and resampling of scRNA-seq data and use the obtained GRN to cluster cells. The clustering is performed from the expression of genes being part of a minimal dominating set (MDS) of the GRN. An MDS is a smallest set of genes that influence each other gene in the GRN. The argument is that cell types could be characterized by these genes only.

The pipeline has been implemented in an R script available online, and evaluated on different datasets. It should be noted that the overall approach is not fully automated: for instance, the number of clusters and assignments of cell types involves manual assessment.

The paper is well written and organized, and figures are clear.

The identification of cellular types and GRN from scRNA-seq data is a very relevant topic, and the proposed methodology seems original and relevant.

Nevertheless, there are a couple of points that could be clarified:

- MDSs are not necessarily unique, which may be worth stating in the paper. It would be insightful to assess the robustness of the classification across equivalent MDSs, and almost-minimal DSs.

- The validation of the clustering might be strengthened using CITE-seq datasets, by comparing the clustering with BootCellNet against the clustering with ADTs

- It is not clear how automated is the cell type label assignment. In the text, this seems a rather expertise-driven labeling. Is part of the cell type identification implemented in the R script?

**Have the authors made all data and (if applicable) computational code underlying the findings in their manuscript fully available?**

Reviewer #1: Yes

Reviewer #2: **No: **Data and scripts are available in the github repository, but when running the scripts, a file not found is thrown. I guess some files are still missing.

Reviewer #3: Yes

PLOS authors have the option to publish the peer review history of their article (what does this mean?). If published, this will include your full peer review and any attached files.

Reviewer #1: **Yes: **Henrique De Assis L Ribeiro

Reviewer #2: No

Reviewer #3: No
---

## [Decision Letter · Decision Letter 1]

11 Sep 2024

Dear Dr Kumagai,

We are pleased to inform you that your manuscript 'BootCellNet, a resampling-based procedure, promotes unsupervised identification of cell populations via robust inference of gene regulatory networks.' has been provisionally accepted for publication in PLOS Computational Biology.

Best regards,

Denis Thieffry, PhD

Academic Editor

PLOS Computational Biology

Pedro Mendes

Section Editor

PLOS Computational Biology

Reviewer's Responses to Questions

**Comments to the Authors:**

Reviewer #1: I am satisfied with how the authors addressed the questions raised by the reviewers; I liked that the results made by visual inspection in Figure 5 were repeated and validated with the automated method (Figure 9).

My only point is in paragraph 2 of "Identification of characteristic cell types in COVID-19 patients by BootCellNet2" the sentence "Compared to the clusters in the original report (...), the clustering by BootCellNet2 has a smaller number of clusters ..." was confusing. If I understood correctly, I would write something like, "Compared to the original report (...), BootCellNet2 found a smaller number of clusters ..." On this line of thought, you say that the original report found 32 clusters, and BootCellNet2 found 17, but at some point, you talk about cluster 34. "Clusters 1, 34, and 8 have high expression of FOS, IRF1, and NFKB2, ..." Maybe you meant clusters 1, 3, 4, and 8.

Reviewer #2: The authors addressed the main issues reported in the initial review.

The manuscript has been improved and the results are now more clearly presented.

The software is also easier to install and the github repository includes the necessary files for the tutorial analysis.

Regarding the methodology, the software is now capable of identifying the cell clusters automatically, using a p-value. This is also better described in a new section in the manuscript.

I have no further comments and I recommend the publication of the manuscript.

Reviewer #3: The revision brings welcomed methodological improvements and additional explanations.

**Have the authors made all data and (if applicable) computational code underlying the findings in their manuscript fully available?**

Reviewer #1: None

Reviewer #2: None

Reviewer #3: Yes

PLOS authors have the option to publish the peer review history of their article (what does this mean?). If published, this will include your full peer review and any attached files.

Reviewer #1: No

Reviewer #2: No

Reviewer #3: No

---

## [Editor Report · Acceptance letter]

25 Sep 2024

PCOMPBIOL-D-24-00282R1 

BootCellNet, a resampling-based procedure, promotes unsupervised identification of cell populations via robust inference of gene regulatory networks.

Dear Dr Kumagai,

I am pleased to inform you that your manuscript has been formally accepted for publication in PLOS Computational Biology. Your manuscript is now with our production department and you will be notified of the publication date in due course.

With kind regards,

Anita Estes
